# Caveolin 1 Regulates the Tight Junctions between Sertoli Cells and Promotes the Integrity of Blood–Testis Barrier in Yak via the FAK/ERK Signaling Pathway

**DOI:** 10.3390/ani14020183

**Published:** 2024-01-05

**Authors:** Qiu Yan, Tianan Li, Yong Zhang, Xingxu Zhao, Qi Wang, Ligang Yuan

**Affiliations:** 1College of Veterinary Medicine, Gansu Agriculture University, Lanzhou 730070, China; 18776410232@163.com (Q.Y.); 17899315032@163.com (T.L.); zhangyong@gsau.edu.cn (Y.Z.); zhaoxx@gsau.edu.cn (X.Z.); wangqi@gsau.edu.cn (Q.W.); 2Gansu Key Laboratory of Animal Generational Physiology and Reproductive Regulation, Lanzhou 730070, China; 3College of Life Science and Technology, Gansu Agriculture University, Lanzhou 730070, China

**Keywords:** yak, RAN-seq, cryptorchidism, blood–testis barrier, tight junctions

## Abstract

**Simple Summary:**

Cryptorchidism is believed to be one of the leading causes of infertility in male yaks. In this study, we compared the morphology of the normal testis of the yak with that of the cryptorchidism, and found dysplasia of the seminiferous tubules, impaired tightness of the Sertoli cells, and a disruption of the integrity of the blood–testis barrier (BTB) in the cryptorchidism. Based on RNA-seq, bioinformatics analysis and biological experiments, CAV1 up-regulates or down-regulates the expression levels of ZO-1, occludin and claudin-11 via the FAK/ERK pathway in vitro cell model assays. Our results reveal a novel mechanism by which CAV1 modulates tight junctions and BTB, suggesting that CAV1 may be involved in the regulation of the occurrence of yak cryptorchidism.

**Abstract:**

Yaks, a valuable livestock species endemic to China’s Tibetan plateau, have a low reproductive rate. Cryptorchidism is believed to be one of the leading causes of infertility in male yaks. In this study, we compared the morphology of the normal testis of the yak with that of the cryptorchidism, and found dysplasia of the seminiferous tubules, impaired tightness of the Sertoli cells, and a disruption of the integrity of the blood–testis barrier (BTB) in the cryptorchidism. Previous studies have shown that CAV1 significantly contributes to the regulation of cell tight junctions and spermatogenesis. Therefore, we hypothesize that CAV1 may play a regulatory role in tight junctions and BTB in Yaks Sertoli cells, thereby influencing the development of cryptorchidism. Additional analysis using immunofluorescence, qRT-PCR, and Western blotting confirmed that CAV1 expression is up-regulated in yak cryptorchidism. CAV1 over-expression plasmids and small RNA interference sequences were then transfected in vitro into yak Sertoli cells. It was furthermore found that CAV1 has a positive regulatory effect on tight junctions and BTB integrity, and that this regulatory effect is achieved through the FAK/ERK signaling pathway. Taken together, our findings, the first application of CAV1 to yak cryptorchidism, provide new insights into the molecular mechanisms of cell tight junctions and BTB. This paper suggests that CAV1 could be used as a potential therapeutic target for yak cryptorchidism and may provide insight for future investigations into the occurrence of cryptorchidism, the maintenance of a normal physiological environment for spermatogenesis and male reproductive physiology in the yak.

## 1. Introduction

The most crucial characteristic in the survival of living species is reproduction. Yaks are a unique bovine species endemic to high-altitude regions. Cryptorchidism is a common disease of the reproductive system and one of the main causes of infertility in male yaks [1]. 

Cryptorchidism, a familiar disease of the male reproductive system, is one of the main causes of male yak sterility. Statistics reported that cryptorchidism occurs in about 0.17% of bulls, of which 34% testes were located in the abdominal cavity [2,3]. In general, the production and secretion of testosterone in the Leydig cells of cryptorchid bulls is mildly impaired, and as a result, cryptorchid bulls are shunned in the market due to their greatly reduced value because of reduced productivity and feeding management issues [4,5]. However, studies of cryptorchid infertility have focused on humans and rodents, and there has been a lack of attention to yak cryptorchidism [6]. Cryptorchidism induces disturbances in the seminiferous epithelium and interstitial tissue of the ectopic testes; further, it causes changes in testicular capillary angiogenesis or degeneration and a thickening of the endovascular cortex, and ultimately impedes the material exchange [7]. Cryptorchidism is constantly associated with decreased sperm motility, decreased sperm count, decreased sperm mass, and abnormal sperm morphology [8]. It is now recognized that cryptorchidism is caused by the failure of testicular descent to the scrotum, which is located in the abdominal cavity and subcutaneous fat layer, resulting in hyperthermia, ischemia, hypoxia and the thickening of the capillary endothelium [9]. These changes in the internal environment hinder the exchange of testosterone and nutrients between cells, and disrupt the tight junctions between Sertoli cells (SCs) and spermatogenic cells and between SCs and Sertoli cells. Importantly, SCs are an important component of the blood–testis barrier (BTB) and the microenvironment of spermatogonial stem cells, providing supports and nutrition for the growth of germ cells throughout spermatogenesis. Ultimately, this change further impairs cell adhesion and leads to BTB destruction [10].

The function of the BTB is to sequester germ cells and lymphatic systems, together with local immune suppression [11]. BTB acts not only as an anatomical barrier, but also as an immunological barrier to provide a complex physiological environment for spermatogenesis. This structure includes tight junctions (TJs) between adjacent SCs, gap junctions, desmosomes and ectoplasmic specialization (ES) [12,13,14]. TJs are an important structure that constitutes the BTB, which is mainly composed of membrane integrity proteins, namely claudins, occludin and zonula occludens (ZOs), which are members of the surrounding protein family [15]. The maintenance of the normal function of the BTB depends primarily on the expression levels of these proteins, and the absence of these proteins continuously increases the body’s autoimmune response to spermatogenesis, which is accompanied by decreased spermatogenesis [16]. For example, when the functional status of occludin in testicular SCs is disturbed, it can hinder the normal occurrence of sperm and lead to infertility [17].

Caveolae, tiny cavities around 50–100 nm, were first described in the 1950s. They are flask-shaped invaginations of plasma membranes present on a large number of mammalian cells and have been implicated in a wide variety of cellular events, such as endocytosis, pinocytosis, cholesterol trafficking, and cellular centers that are essential in coordinating signaling events [18]. Caveolae are coated by the caveolin protein family, and there are three isoforms of the caveolin family: caveolin-1 (CAV1), CAV2 and CAV3 in mammalian cells [19]. Among these, CAV1 is a membrane-integrating protein whose presence has been demonstrated in most cell types, including epithelial cells, endothelial cells, fibroblasts, smooth muscle cells and adipocytes [20]. CAV1 contains a highly conserved caveolin-scaffolding domain (CSD), located in Amino acids 82-101 in the N-terminus immediately adjacent to the central hydrophobic region, which mediates the CAV1 interaction with multiple signaling molecules, such as the SRC family tyrosine kinases, growth factor receptors, endothelial nitric oxide synthase (eNOS) and G proteins [21,22]. Hence, it is generally believed that CAV1 is not only a structural protein, but also participates in a wide range of cellular physiological processes because of its characteristics and properties. These include vesicle transport, signal transduction and differentiation regulation, which participate in the occurrence, development and prognosis of related diseases [23].

At present, a growing number of studies have reported that CAV1 can regulate TJ-related proteins. Studies have shown that the disruption of the epithelial barrier may be related to the reduction of CAVI expression [24]. Moreover, CAV1-dependent and occluded endocytosis is essential for in vivo immune-mediated TJ regulation [25]. At present, we know that the normal expression of tight junction-associated proteins is essential for the maintenance of BTB integrity; however, the molecular mechanisms underlying TJs and BTB injury remain unclear and are in urgent need of further elucidation. In this paper, we attempt to investigate whether CAV1 may further influence a range of male reproductive events in the yak, such as spermatogenesis, infertility and cryptorchidism, by modulating the expression of TJ-related proteins in the SCs and the integrality of BTB.

## 2. Materials and Methods

### 2.1. Animals and Sample Collection

Testis (*n* = 3) and cryptorchidism (*n* = 3) were collected from a local slaughterhouse in Tianzhu County, Gansu Province, China. The testis tissue was washed with phosphate-buffered saline (PBS) containing penicillin and streptomycin. Normal testis and cryptorchidism tissues were then decapsulated and separated into small pieces. One portion was fixed in 4% neutral paraformaldehyde for histopathological and pathological analysis. Meanwhile, another part was frozen in liquid nitrogen until it was returned to the laboratory and kept at −80 °C. For testicular tissue used for subsequent primary cell culture, care was taken to maintain the integrity of the testes after harvesting the entire testis. After 1 min of ice-free sterile PBS containing penicillin and streptomycin flushing, the testicular tissue was immersed in sterile PBS under cryogenic conditions and transferred to the laboratory as soon as possible for subsequent processing.

### 2.2. Identification via RNA-seq and Bioinformatics Analysis of miRNAs and mRNAs in Yak Testis and Cryptorchidism

The mRNA abundances of differentially expressed mRNAs and miRNAs were screened and quantified by RSEM software (RESM v1.1.17). We then use the FPMK method to normalize the expression level for each sample [26]. The miRNA expression level from each sample was calculated and normalized to TPM (TPM = actual miRNA count/total count of clean reads × 10^6^). The analyses of differentially expressed (DE) miRNAs and mRNA between the two groups were accomplished by the edgeR package (http://www.bioconductor.org/packages/release/bioc/html/edgeR.html, accessed on 13 January 2022). DE mRNAs and miRNA were distinguished with an absolute fold change >1, FDR < 0.05 and *p* < 0.05. Gene Ontology (GO) annotation and Kyoto Encyclopedia of Genes and Genomes (KEGG) enrichment analysis were performed for all DE mRNAs and miRNAs using the GO database (http://www.geneontology.org/, accessed on 14 January 2022) and KEGG database (http://www.genome.jp/kegg/pathway.html, accessed on 14 January 2022). respectively.

### 2.3. Hematoxylin–Eosin Staining

The testis tissue was fixed in 4% neutral paraformaldehyde. Washing, dehydration and paraffin embedding were performed according to routine histopathological protocols. Hematoxylin and eosin (H&E) staining was performed using a Hematoxylin–Eosin/HE Staining Kit (G1120, Solarbio, Beijing, China), which can be summed up by dewaxing with xylene, dehydrating with alcohol gradient, staining with hematoxylin and eosin, dehydrating again, dewaxing with xylene, and finally sealing with neutral balsam according to the instructions of manufacturer. And, the slides were examined under a light microscope equipped with a digital camera (Olympus, Tokyo, Japan).

### 2.4. Plasmid Construction, Cloning and Double Digests

Total RNA was isolated from SCs using Triquick Reagent (R1100, Solarbio). The RNA must immediately reverse-transcribe to cDNA under the low temperatures using an Evo M-MLV RT Kit (AG11728, Accurate Biology, Changsha, China) in terms of the manufacturer’s instructions. The coding sequence of yak CAV1 (GenBank ID: NW_005394073.1) was amplified by a polymerase chain reaction (PCR; Bio-Rad, Hercules, CA, USA) using Premix Taq DNA Polymerase (TaKaRa, Terra Bella Ave., Moutain View, CA, USA) and cloned into the p-UC-57 vector. DNA fragments encoding the complete yak CAV1 coding sequence were amplified by PCR using Premix Taq DNA Polymerase. The product was then ligated into the pIRES2-EGFP (p-EGFP) vector to construct the pIRES2-EGFP-CAV1 (p-EGFP-CAV1) expression vector. The flanking *EcoR* I (R0101V, New England Biolabs, Ipswich, MA, USA) and *BamH* I (R0136V, New England Biolabs) restriction sites were created and the DNA fragments were verified by agarose gel electrophoresis.

### 2.5. Cell Primary Culture and Identification

SCs were isolated, as recently described, with some alterations [26,27]. In brief, SCs were isolated by a mixed digestion with trypsin and collagenase IV. Briefly, under aseptic conditions, testes were rinsed three times with PBS (containing 100 U/mL penicillin and 100 mg/mL streptomycin). After the removal of the tunica albuginea, the seminiferous tubules were minced and digested with type-IV collagenase (1 mg/mL) mixed with 0.25% trypsin (volume ratio 1:1) for about 35 min at 37 °C. When the tissue suspension was observed to be stratified and the upper layer was clarified, the digestion was terminated with Dulbecco’s Modified Eagle’s medium/high glucose (DMEM; 01-052-1A, Biological Industries, Kibbutz Beit Haemek, Israel) containing 10% fetal bovine serum (10099141C, Gibco, Grand Island, NY, USA), and the digested cells were filtered through a 200-mesh screen from the suspension. After centrifugation at 1000 rpm (15 min) and PBS washes (two times), the SCs containing sperm were cultured with DMEM/high glucose supplemented with 10% fetal bovine serum containing streptomycin (50 μg/mL) and penicillin (50 IU/mL) at 37 °C, in 5% CO_2_ atmosphere. After 24–48 h of culture, the medium was replaced with fresh medium to remove suspended germ cells. After 18 h of continued culture, the supernatant was discarded and 0.05% trypsin was added to transiently digest the cells before fresh medium was added. When the cell reaches 80% density, the above procedure is repeated three or four times to obtain pure SCs. In order to verify the purity of isolated SCs, SCs were identified via immunofluorescence using the wilm tumor gene1 (WT1) protein, which is a specific marker of SCs.

### 2.6. Cell Transfection

In the overexpression group, cells were plated in a 35-mm dish for transfection so that the monolayer cell density reached an optimal 90–95% confluency at the time of transfection. Transfected SCs with an overexpression of plasmid-encoding yak CAV1 were fused to the fluorescent reporter protein p-EGFP (p-EGFP-CAV1). To be specific, DMEM (1 mL) or complete fresh medium was added to each well 30–60 min before transfection. For SCs, the optimal ratio of PolyJet^TM^ (μL): DNA (μg) is around 3:1. For each 35 mm dish, 1 μg of DNA was diluted in 50 μL of DMEM and gently pipetted up and down or vortexed briefly to mix. Three microliters of polyJet reagent were diluted in 50 μL of DMEM. Subsequently, diluted PolyJet^TM^ reagent (SignaGen, Frederick, MD, USA) was added to the diluted DNA solution all at once and immediately pipetted up and down 3–4 times or vortexed briefly to mix, and incubated for 15–30 min at room temperature to allow PolyJet^TM^/DNA complexes to form. 100 μL of PolyJet^TM^/DNA mixture were added drop-wise onto the medium in each well, and the mixture was homogenized by gently swirling the plate. It was put into a cell incubator at 37 °C. The PolyJet^TM^/DNA mixture-containing medium was removed and replaced with a completely fresh medium after 12–16 h, and the transfection efficiency was determined 72 h after the transfection of the overexpression vector.

Cells were plated in a 35 mm dish prior to transfection to ensure that the monolayer cell density reaches an optimal confluency of 30–50%. The siRNA for yak CAV1 was designed and synthesized by the Ribo-Bio Company (Guangzhou, China). Lipofectamine^@^2000 Reagent (Thermo Fisher Scientific, Carlsbad, CA, USA) was used to transfer siRNA according to the manufacturer’s instructions and previous approaches [28].

### 2.7. Quantitative Reverse Transcription-PCR

The total RNA from tissues and cells was extracted using Triquick Reagent and immediately reverse-transcribed to cDNA using an Evo M-MLV RT Kit with gDNA Clean for qRT-PCR, according to the manufacturer’s instructions. qRT-PCR was performed using 2 × SYBR Green qPCR Master Mix (B21202, Bimake, Houston, TX, USA) according to the manufacturer’s instructions and evaluated using the LightCycler 96 real-time system (Roche, Basel, Switzerland). The two-step amplification program is as follows: 95 °C for 600 s and 40 two-step amplification cycles of 95 °C for 15 s and 60 °C for 45 s. The housekeeping gene *GAPDH/U6* was used as an intra-tissue control. Table 1 lists the sequences of target genes whose relative expression changes were computed by the (2^−ΔΔCT^) approach.

### 2.8. Immunofluorescence and Immunohistochemistry

In addition to identifying yak SCs, immunofluorescence and immunohistochemistry were used to detect CAV1 expression in yak SCs and testis tissues. The immunofluorescence methods used for cells and tissues were slightly different. The cells were cultured for 48 h, and a fluorescence microscope (Olympus, Tokyo, Japan) was used to observe the transfection efficiency. The cells were fixed in 4% paraformaldehyde for at least 20 min first, and then washed with 4 °C ice-cold PBS three times, treated with 0.1% Triton X-100 for 30 min at room temperature, washed three times, and then incubated with 5% bovine serum albumin (A8010, Solarbio, Beijing, China) for 30 min. Cells were then incubated with mouse monoclonal antibody for CAV1 (1:300) at 4 °C overnight. Goat anti-mouse IgG H&L (Alexa Fluor^®^ 488) secondary antibody (ab150113, 1:300, Abcam, Cambridge, UK) was used at 37 °C for 1 h in the dark. Cell nuclei were stained with 1 μg/mL 4′,6-diamidino-2-phenylindole (D9542, Sigma-Aldrich, St. Louis, MO, USA). Digital images were acquired using an Olympus DP73 optical microscope (Olympus, Tokyo, Japan). Sections (4 μm) were mounted onto gelatin/poly-L-lysine-coated glass slides and dried in an incubator at 60 °C for 2 h, and we referred to previous literature for the rest of the methodology. The sections were observed and photographed using an Olympus-DP73 optical microscope after using an antifade mounting medium (P0126, Beyotime, Shanghai, China).

For immunohistochemistry, the paraffin sections were de-paraffinized by graded alcohols and xylene, rehydrated, and washed thrice with PBS. The antigens in specimens were retrieved with a sodium citrate buffer solution (G2700, Solarbio). The immunohistochemistry staining was carried out using a Histone H3 Ready-To-Use IHC Kit (IHC0111, Bioss, Beijing, China), according to the instructions. The sections were incubated with rabbit primary antibody (CAV1,1:100; ZO-1,1:100; occludin, 1:100; claudin-11, 1:75) overnight at 4 °C in a wet box. Next, positive signals were visualized by a 30-diaminobenzidine (DAB) kit (C-0003, Bioss). The representative images were captured using an optical microscope (Olympus).

### 2.9. Western Blotting

The total protein was isolated from yak testes using a cell lysis buffer (R0010, Solarbio). A loading buffer was added to the protein samples, and the samples were denatured in a water bath at 100 °C for 10 min. Samples containing the same amount of protein (40 μg) were separated with 12% SDS-polyacrylamide gels. The blots were electro-transferred onto a PVDF membrane (IPVH00010, Millipore, Billerica, MA, USA), then blocked for 2 h in Tris-HCl buffer containing 5% (*w*/*v*) nonfat milk powder. The membranes were incubated overnight at 4 °C with primary antibodies. The antibodies used are briefly described in this section. They include the mouse monoclonal antibody for CAV1 (66067-1-Ig, 1:2000, Proteintech, Wuhan, China), mouse monoclonal antibody for FAK (66258-1-Ig, 1:1500, Proteintech), rabbit polyclonal antibody for JNK1 (T55490, 1:1500, Abmart, Shanghai, China), rabbit monoclonal antibody for ERK1 (T55561, 1:1000, Abmart), rabbit monoclonal antibody for phospho-SRC (ab185617, 1:2000, Abcam, Cambridge, UK), rabbit monoclonal antibody for SRC (GB112343-100, 1:500, Servicebio, Wuhan, China), rabbit monoclonal antibody for ERK1 (TA0155S, 1:2000, Abmart), rabbit polyclonal antibody for phosphor-ERK1 (TA1015S, 1:1000, Abmart), rabbit polyclonal antibody for ZO-1 (TA5145, 1:1000, Abmart), mouse polyclonal antibody for occludin (66378-1-Ig, 1:500, Proteintech), rabbit polyclonal antibody for claudin-11 (TA5364S, 1:800, Abmart) and mouse monoclonal antibody for GAPDH (60004-1-Ig, 1:5000, Proteintech). After washing, membranes were incubated with horseradish peroxidase (HRP)-conjugated goat anti-rabbit (bs-0295G-HRP, Bioss) or goat anti-mouse secondary antibody (bs-0368G-HRP, Bioss) at a dilution of 1:5000. Immunocomplexes were detected using an enhanced chemiluminescence solution (Abnova, Taipei, Taiwan, China), and signals were quantified using ImageJ (V 1.8.0).

### 2.10. Statistical Analysis

Statistical analyses were performed using SPSS (version 22.0; SPSS Inc., Chicago, IL, USA). Data were expressed as the mean ± SD using Prism version 5.0 (GraphPad Software Inc., La Jolla, CA, USA). Data were analyzed using Student’s *t*-test (between two groups) or one-way analysis of variance (within multiple groups). Differences between the two groups are marked by ** p* < 0.05 and *** p* < 0.01.

## 3. Results

### 3.1. Cell TJs and BTB Were Detected in yak Testis and Cryptorchidism

The mRNA levels of *ZO-1*, *occludin* and *claudin-11* were detected via qRT-PCR, and protein levels were analyzed by Western blotting and IHC in the testis and cryptorchidism. H&E staining revealed that, in normal testes, the seminiferous tubules are well developed; the blood vessels are distributed in the interstitial regions of adjacent seminiferous tubules; and the interstitial cells are numerous and large, with abundant cytoplasm, rounded or oval-shaped nuclei, and prominent nuclei. The spermatogenic cells are arranged in multiple layers with a long, spindle-like shape. Peritubular myoid cells are distributed around the lamina propria of the seminiferous tubule. Scattered primary spermatophores and sperm are observed in the lumen. In the cryptorchidism, however, the stroma is loose, the LCs are reduced, the tissue structure is unclear, and the seminiferous tubules are poorly developed and appear to contract. The number of spermatogenic epithelial cells in the layer is significantly reduced and the number of spermatogenic cells is small, with some of them shed in the lumen. With all samples producing a clear signal for *GAPDH* mRNA, which acted as a housekeeping gene control, the PCR and Western blotting results suggested that ZO-1 and occludin mRNA and protein levels were both significantly down-regulated in cryptorchidism compared to those in the control (testis) (*p* < 0.01) (Figure 1B,C). The results of IHC show that the ZO-1, occludin and claudin-11 protein exhibited similar expression profiles in testis and cryptorchidism with localization in various germ cells, SCs and Leydig cells (LCs) (Figure 1D–F).

### 3.2. Identification and Functional Enrichment Analysis of Differentially Expressed Genes in the Normal Testis and Cryptorchidism of Yak

We carried out mRNA-seq analysis from testis (*n* = 3) and cryptorchidism (*n* = 3). Here, we identified a total of 21,620 mRNAs (19,262 genes), of which 683 genes were unknown Appendix A). Based on |fold change| > 1, FDR < 0.05, a total of 575 differential genes were identified; 497 were up-regulated and 78 were down-regulated (Figure 2A) (Appendix A). The GO and KEGG enrichment analysis showed that most DEGs genes were mainly involved in biological processes, molecular function and cellular component, which focus on metabolic pathways, the AMPK signaling pathway, Alzheimer’s disease, endocytosis, the PI3K/Akt/mTOR signaling pathway and focal adhesion (Figure 2B,C). Herein, for further screening and analysis of potential target-related genes, the GO enrichment analysis suggested that most DEGs genes were mainly involved in response to stimulus, binding and metabolism; the regulation of a biological process; and the metabolic process (Figure 2D,E). KEGG suggest that CAV1 most focuses on focal adhesion, proteoglycans in cancer, fluid shear stress and atherosclerosis, viral myocarditis and the bacterial invasion of epithelial cells (Figure 2F).

### 3.3. Identifying Differentially Expressed microRNA of Small RNA Sequencing

According to the homology of miRNAs and the highly conservative unique read long, using small RNA sequencing through mirdeep2 software (v0.1.3) predicted microRNAs (Figure 3). For the first time, fragment lengths of microRNAs from yak testis and cryptorchidism were analyzed. We found a peak in the length distribution, with 90% of the detected small RNA being microRNAs of the length 20–24 bp (Figure 3A). A total of 1454 microRNAs were detected in the six yak testis samples (Figure 3B) (Appendix A). The types and percentages of small RNA were further identified, and statistics of all samples are shown in Figure 3C,D. There are 67.81% of microRNAs known, and 32.19% of microRNAs were unknown in the miRbase database. After intelligently using a stringent filtering criterion (fold change > 1 and *p* < 0.05) by pairwise intergroup alignment, 103 microRNAs were identified, 48.54% of which were up-regulated (a total of 50 miRNAs: 39 known; 11 novel), and 51.46% were down-regulated (a total of 53 miRNAs: 33 known; 20 novel) (Figure 3E) (Appendix A). In total, we found that TGFB1 and CAV1 mainly presented a negatively integrated analysis-correlated expression pattern with 13 miRNAs based on an miRNA–mRNA co-expression network analysis (Figure 3F) (Appendix A). Appendix A see Appendix A.

### 3.4. Validation of DERs and DEGs

Real-time PCR was performed to validate the results of the eight different expression miRNAs (DERs) and eight different expression genes (DEGs). The PCR results suggested that all DERs and DEGs were differentially expressed in the testis and cryptorchidism, four DERs were significantly up-regulated and four DERs were significantly down-regulated; this trend was in accordance with the predicted results of miRNA sequencing (Figure 4A). Eight DEGs were significantly up-regulated at the expression level (Figure 4B); this trend was in accordance with the predicted results of mRNA sequencing. To verify the abundance of variation in protein levels, we choose two potential target proteins, TGFB1 and CAV1, for Western blotting assays. Target proteins were differentially expressed in testis and cryptorchidism, and the expression tendency and expression levels was similar to that of mRNA of CAV1. However, the TGFB1 protein level was not clearly different (Figure 4C). We further verified the location and distribution of the target protein CAV1 in testis and cryptorchidism, and the results showed that the CAV1 protein had a strong positive expression in LCs and SCs, and a weak positive expression in sperm cells. In addition, CAV1 was expressed in peritubular cells and spermatogonia. And, the positive expression of CAV1 in cryptorchidism was stronger than that in testis (Figure 4D,E).

### 3.5. The Efficiency of p-EGFP-CAV1 Was Detected in Yak Sertoli Cells

From the IF results, the nuclei are stained blue and WT1 is stained red, and the stained results indicate the high purity of the separation and purification of SCs, which can be used in subsequent cell transfection (Figure 5A). The IF analysis showed that when yak SCs were transfected with the fluorescent reporter protein p-EGFP (p-EGFP-CAV1), the green fluorescence intensity of the vector p-EGFP reaches its optimal level 72 h after the in vitro transfection, implying that the transfection efficiency is enhanced at this time (Figure 5B). qRT-PCR and Western blotting analysis were performed to determine the CAV1 mRNA and protein levels in the transfected cells, respectively. Compared with the p-EGFP (control), significantly higher CAV1 mRNA and protein levels were detected after transfection (Figure 5C,D).

### 3.6. Detection of Silencing Efficiency of the CAV1 Gene

To further examine the function of CAV1, we designed two siRNA-CAV1 sequences and a negative siRNA-NC (control) and transfected them into Sertoli cells for 72 h. qRT-PCR results indicated that two target sequences had silencing efficiency, and siRNA-CAV1-1 was the most efficient construct of the two tested, significantly reduced CAV1 mRNA levels compared with siRNA-NC expressing cells (*p* < 0.01) (Figure 6A). The Western blotting of CAV1 protein levels showed a comparable reduction induced by siRNA-CAV1-1 compared with siRNA-NC, as expected (Figure 6B).

### 3.7. Up-Regulated CAV1 Promoted TJs and BTBs via FAK/ERK Pathway of Sertoli Cells

To further investigate the regulated function of CAV1 on TJs and BTB in yak SCs, the ZO-1, occludin and claudin-11were detected by the qRT-PCR and Western blotting analysis when the expression level of CAV1 was overexpressed (Figure 6). The results showed that the expression of ZO-1, occludin and claudin-11 mRNA and protein were significantly up-regulated in yak SCs, compared with the control (*p* < 0.01). In order to explore the molecular mechanism of CAV1 regulating cells’ adhesion and tight junctions, we detected its downstream-related molecules, and the results showed that the mRNA levels of FAK, SRC, JNK1 and ERK1 were significantly up-regulated (*p* < 0.05 or *p* < 0.01) (Figure 7A–G). The protein levels of FAK, JNK1 and ERK1 were significantly increased; however, the SRC protein levels were not significantly different. To determine the effect of up-regulation CAV1 induced SRC and ERK1 phosphorylation, we assayed SRC and ERK1 phosphorylation using Western blotting, the Up-regulated CAV1 significantly increased the levels of phosphorylated SRC and ERK1 (*p* < 0.05) (Figure 7H).

### 3.8. Down-Regulating CAV1 Suppression of TJs and BTB via FAK/ERK Pathway of Sertoli Cells

Expression levels of ZO-1, occuldin and claudin-11 were measured by qRT-PCR and Western blotting. mRNA and protein expression levels of ZO-1, occuldin and claudin-11 in CAV1 knock-down SCs were significantly decreased compared with controls (*p* < 0.01) (Figure 8E–G). To verify whether the decreased FAK/ERK expression is CAV1-dependent or not, the expression levels of related genes and proteins in the downstream signaling pathway were examined for yak SCs. The qPCR results revealed that the mRNA expression level of *FAK, SRC, JNK1* and *ERK1* was significantly down-regulated compared with controls (*p* < 0.05 or *p* < 0.01) (Figure 8A–D), and the protein levels of FAK, JNK1 and ERK1 were decreased. Surprisingly, silencing CAV1 significantly decreased the levels of phosphorylated SRC and ERK1 (Figure 8H).

## 4. Discussion

A healthy testis is the primary factor in normal male reproductive function, and cryptorchidism increases the risk of male infertility in the yak. SCs are the key cells in the testis, which are also known as ‘nurse’ cells, and provide nutrients, paracrine factors, cytokines, and other biomolecules to support germ cell development [29]. Spermatogenesis is a particularly complex physiological process supported by the intricate crosstalk between SCs [30,31]. The most important of these is that SCs establish the BTB [32]. In BTB, TJs coexist and cofunction with ectoplasmic specializations, desmosomes and gap junctions to create a unique microenvironment for the completion of meiosis and the subsequent development of spermatids [11]. The changes in the relevant molecules of the BTB need to be investigated during the dynamic spermatogenesis [11,33]. In this study, we focus on those changes concerned with the key proteins of TJs [12]. TJs formed by adjacent SCs mainly comprise zonula occludens-1 (ZO-1), occludin, claudin-11 and the junction adhesion molecule [34]. When SCs are stressed, claudin-11 and occludin were decreased in both mRNA expression and protein level [35]. It is noteworthy that the communication between SCs is greatly influenced by the relative expression of the TJs protein [36]. At present, although a large number of researchers are devoted to the study of the regulatory mechanisms affecting the expression of proteins associated with cell junctions [37,38], in species such as the yak, studies are rare. In this study, we aim to explore the molecular regulatory mechanisms of TJ proteins in yak SCs. 

CAV1 is well established as a major structural protein of caveolae [39,40]. As a transmembrane protein, both of its terminals are located in the cytoplasm [41]. CAV1 have been implicated in various cellular events such as endocytosis cellular signaling, membrane transport, and immunity, as well as in related diseases [42]. Of course, the trajectory of CAV1 activity is much more than that. CAV1 is also involved in the regulation of cell fates such as cell proliferation, survival and differentiation. For instance, the down-regulation of CAV1 subsequently enhanced the invasion of cancer cells [43]. However, much remains to be discovered about CAV1 structure, formation, regulation and function in different cell types and pathology, particularly in the cells associated with male reproduction.

Based on this literature review, we found that CAV1 plays an important role in TJs and cell adhesion, and is an active regulator, influencing barrier permeability. For example, CAV1 can increase the blood–brain barrier permeability through mediating TJ protein endocytosis and translocation [44]. What is more, hypoxia up-regulates CAV1 transcription and induces the internalization and autophagic degradation of claudin-5, leading to the disruption of the TJs of the blood–brain barrier [45].

Does CAV1 also regulate the TJs of the BTB? In this study, we compared the ultra-structures of normal testes and cryptorchidism, and the results of the H&E showed the presence of severe lesions in the cryptorchid tissue compared to normal testes, including severely shrunken seminiferous tubules, diffuse necrotic degeneration of the germ cells, and damaged TJs between the SCs. And, these changes were associated with cell TJs, which comprise the adhesion of the BTB. With the help of RNA-sequencing, we analyzed and compared the differential miRNAs between normal testis and cryptorchidism in yak. Finally, based on the GO analysis and KEGG analysis, the target protein CAV1, which is associated with TJs, was obtained. The results indicate that CAV1 is up-regulated in the cryptorchidism of yak, and its regulatory mechanism on cell-tight junctions and BTB needs further validation.

Focal adhesion kinase (FAK), a cytoplasmic protein-tyrosine kinase, is a key regulator of cell movement, adhesions, tight junctions, endosomes, and the nucleus [46,47]. Studies have shown that FAK localized to the BTB in virtually all stages of the seminiferous epithelial cycle, which can interfere with the TJs and adhesion between Sertoli cells, destroy BTB and eventually lead to male infertility [48,49]. The integrity of TJs and the regulation of dynamic junction proteins are important for normal spermatogenesis, especially in controlling germ cell migration [50]. ERK is a mitogen activated kinase-like (MAPK) known to regulate FAK [51]. We know that the MAPK/extracellular signal regulated the kinase (ERK) pathway and c-jun N-terminalkinases (JNKs) have modulatory roles on several TJs molecules in the male reproductive system, thus controlling the paracellular permeability of the epithelium [52]. Interestingly, the activation of ERK also functions as a mediator to suppress the claudin-11 and occludin expression in response to hormone stimulation or environmental toxicant exposure [53,54]. What is more, there are studies that have shown that interleukin 6 (IL-6) perturbs the integrity of the BTB, and alters the normal localization and steady-state levels of membrane protein β-catenin and occludin; this regulation was depending on activating ERK pathways [55]. Researchers have confirmed that the activation of p-ERK and p-JNK can protect against testis damage and spermatogenic disorders [56]. Dietary high-fructose can directly or indirectly affect the reproductive function of the testis, while some results suggested that kefir could probably promote testicular function by increasing the integrity of BTB and regulating spermatogenesis. In detail, kefir suppressed the mitogenic signaling cascade and promoted testicular function by decreasing ERK1/2 and p-ERK1/2 levels, but increased JNK1 and p-JNK1 protein levels in testis of high-fructose-fed rats [57]. In addition, the JNK signaling pathway was shown to counteract the destructive effects of the p38 MAPK pathway in the BTB disruption induced by cadmium chloride in rats; this indicated the protective function of JNK pathway in seminiferous epithelium [58]. In summary, the FAK, ERK and JNK et al. pathways all play a critical role in the maintenance of TJs between SCs and BTB completeness. And, changes in the expression of these key proteins can modulate spermatogenesis, sperm function and germ cell apoptosis. This modulation of the reproductive system may have great potential implications for the treatment of male cryptorchidism infertility.

In the present study, we first discovered that the mRNA and protein expression of CAV1 differ between normal testes and cryptorchidism of yaks. To further investigate the function of CAV1, we transfected yak SCs with siRNA sequences and pIRES2-EGFP-CAV1. In the CAV1 knockdown group, the mRNA and protein expression of CAV1, FAK, SRC, JNK1 and ERK1 were reduced, as were the TJs and BTB-related proteins ZO-1, occludin and claudin-11. However, in the CAV1 over-expression group, the mRNA and protein expression trends for these proteins were entirely reversed, with significant increases. In a word, CAV1 can function as a positive regulator to regulate the TJs of SCs during cryptorchidism development via the FAK/ERK signaling pathway (Figure 9). However, the specific role of CAV1 and TJs between SCs in BTB remains to be further investigated. The complex mechanisms of SCs underly their function as “nurse cells”; how to apply this information to clarify the effect of SCs on fertility is still a hot spot that needs continuous attention and exploration.

## 5. Conclusions

In conclusion, we identified the presence of a disruption in cellular TJs and the integrity of BTB, while CAV1 expression is up-regulated in yak cryptorchidism compared to normal testes. And, CAV1 can positively regulate TJs of SCs during cryptorchidism development via the FAK/ERK signaling pathway. This research will be helpful for further understanding the regulatory mechanism of CAV1 in testicular spermatogenesis and the occurrence of cryptorchidism.

## Figures and Tables

**Figure 1 animals-14-00183-f001:**
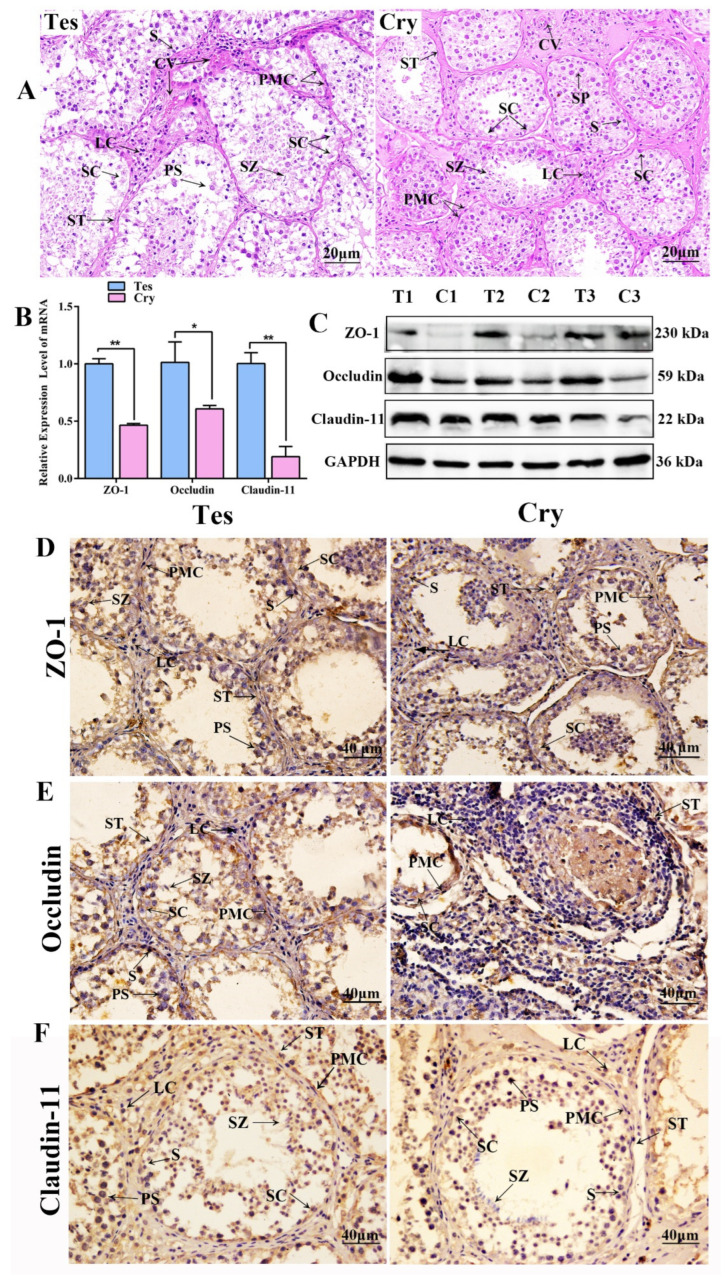
TJs and BTB was detected in yak testis and cryptorchidism. (**A**) Histomorphology was analyzed by H&E staining in yak testis and cryptorchidism. (**B**) The mRNA expression level of *ZO-1*, *occludin* and claudin-11 were detected by qRT-PCR; values represent mean ± SD, *n* = 3. ** p* < 0.05, *** p* < 0.01. (**C**) The protein expression of ZO-1, occludin and claudin-11 were analyzed using Western blotting. (**D**–**F**) Localization of ZO-1, occludin and claudin-11 protein in testis and cryptorchidism were analyzed using immunohistochemistry. PMC: peridubular myoid cells; S: spermatogonium; SZ: spermatozoa; ST: Seminiferous tubule; PS: primary spermatocyte; SC: Sertoli cell; LC: Leydig cell; CV: capillary vessel.

**Figure 2 animals-14-00183-f002:**
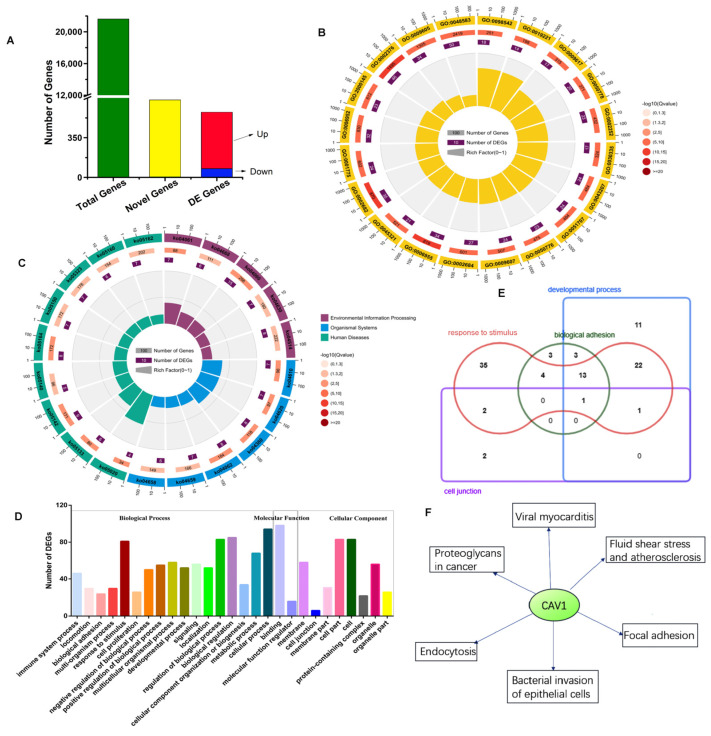
GO and KEGG Analysis of Differently Expressed Target Genes. (**A**) The histogram shows number of mRNA. Green and yellow represent total mRNAs and novel mRNAs; red and blue represent increased and decreased mRNA abundance. (**B**) GO annotation functional classification of differentially expressed genes. (**C**) KEGG pathways with significant enrichment of differentially expressed genes. (**D**–**F**) GO and KEGG analysis difference expression genes for immunity response to stimulus, biological adhesion, development process and cell junction.

**Figure 3 animals-14-00183-f003:**
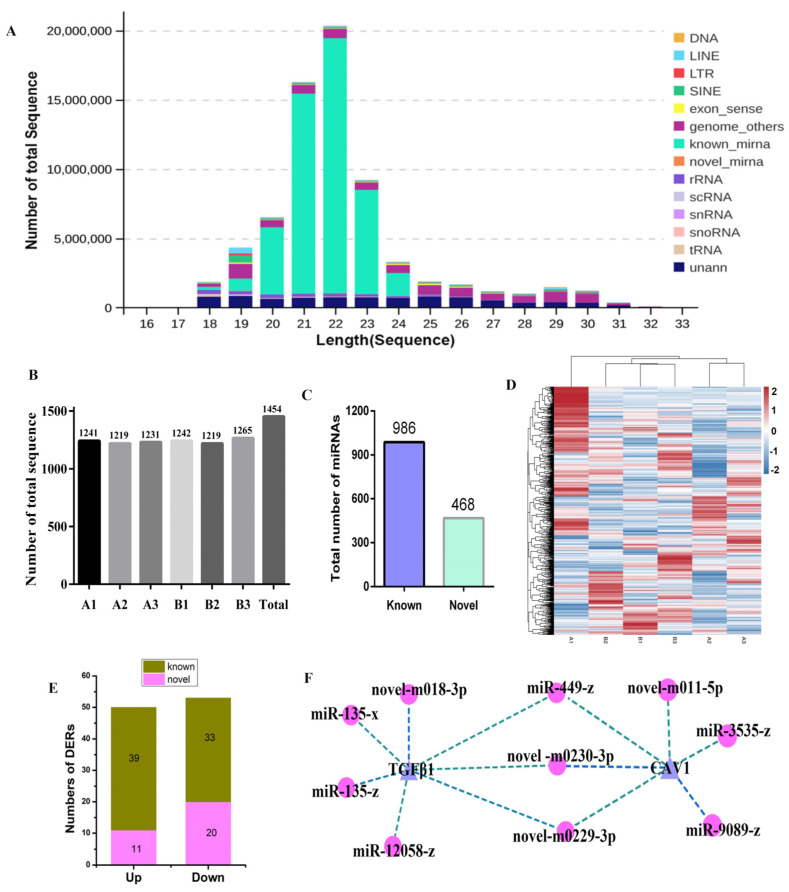
Identifying differentially expressed microRNA of small RNA sequencing. (**A**) The fragment length distribution of microRNAs in yak testis and cryptorchidism. (**B**) The number statistics of miRNA in per samples. (**C**) The total of miRNA in all samples. (**D**) The heatmap of all microRNAs in the two groups. (**E**) The statistical graph of the differentially expressed miRNAs (DERs) between groups. (**F**) TGFB1 and CAV1 were potentially regulated by miRNAs.

**Figure 4 animals-14-00183-f004:**
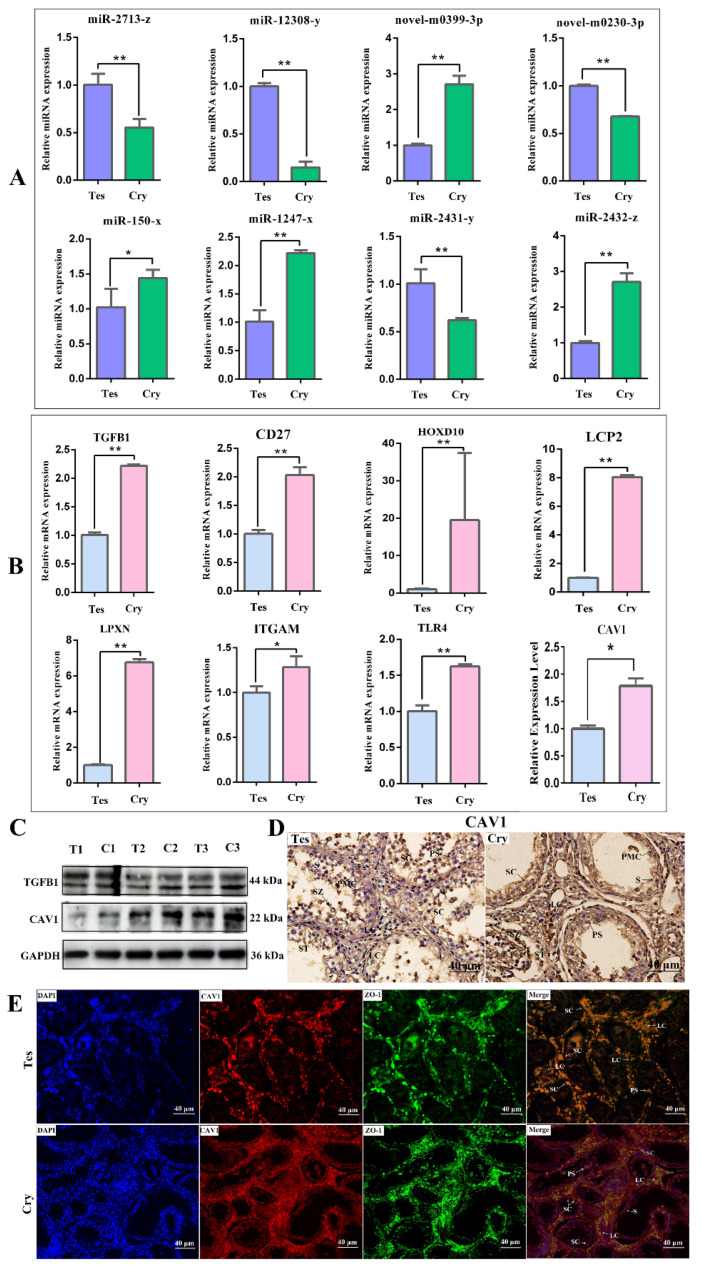
Verification of DE miRNAs and DE mRNAs. (**A**) qPCR analysis for 8 randomly selected miRNAs. Data represent the mean ± SD, *n* = 3, ** p* < 0.05, *** p* < 0.01. (**B**) qPCR analysis for 8 randomly selected mRNAs. Data represent the mean ± SD, *n* = 3, ** p* < 0.05, *** p* < 0.01. (**C**) Expression patterns of TGFB1 and CAV1 proteins by Western blotting analysis, *n* = 3. (**D**) Immunohistochemical stain assay for expression and location of CAV1 in testis and cryptorchidism. (**E**) Intracellular localization analysis of CAV1 protein in Sertoli cells. PMC: peritubular myoid cells; S: spermatogonium; SZ: spermatozoa; ST: seminiferous tubule; PS: primary spermatocyte; SC: Sertoli cell; LC: Leydig cell.

**Figure 5 animals-14-00183-f005:**
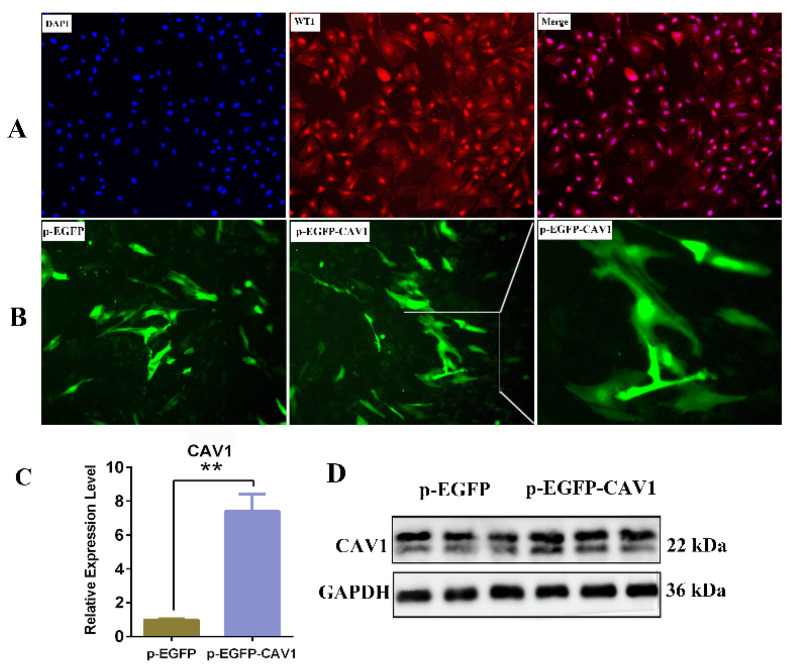
Expression analysis of CAV1 over-expression efficiency in yak Sertoli cells. (**A**) Immunofluorescence staining identified the isolated yak SCs using antibodies against WT1. Original magnification: ×200. (**B**) Fluorescence microscopy of Sertoli cells transfected with an p-EGFP-CAV1 overexpression plasmid or control empty vector (p-EGFP). Cells were examined at 72 h after transfection. Original magnification: ×400. (**C**) qRT-PCR analysis of CAV1 mRNA in Sertoli cells transfected to pIRES2-EGFP-CAV1; values represent mean ± SD, *n* = 3, *** p* < 0.01. (**D**) The protein expression of CAV1 was detected using Western blotting, *n* = 3.

**Figure 6 animals-14-00183-f006:**
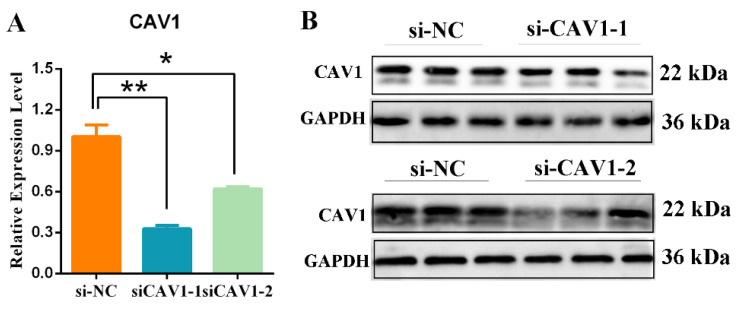
Silencing of CAV1 expression in Sertoli cells. (**A**) qRT-PCR analysis of CAV1 mRNA levels in Sertoli cells transfected for 72 h with a control siRNA or two CAV1-targeting siRNAs. Values represent mean ± SD, *n* = 3, ** p* < 0.05, *** p* < 0.01. (**B**) Western blotting analysis protein level of CAV1 in Sertoli cells after transfection for siRNA-NC, siRNA-CAV1-1 and siRNA-CAV1-2; GAPDH was probed as an internal control, *n* = 3.

**Figure 7 animals-14-00183-f007:**
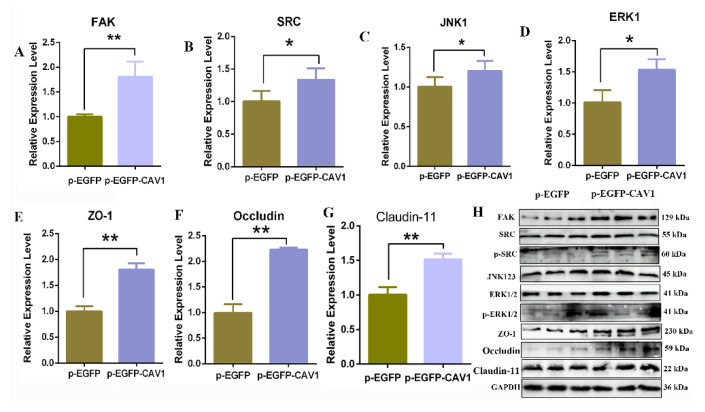
CAV1 promoted TJs and BTB of Sertoli cells. (**A**–**G**) The mRNA expression of FAK, SRC, JNK, ERK, ZO-1, occludin and claudin-11 was detected by qRT-PCR; values represent mean ± SD, *n* = 3. * *p* < 0.05, ** *p* < 0.01. (**H**) The protein expression of FAK, SRC, p-SRC, JNK, ERK1, p-ERK1, ZO-1, occludin and claudin-11 was detected using Western blotting, *n* = 3.

**Figure 8 animals-14-00183-f008:**
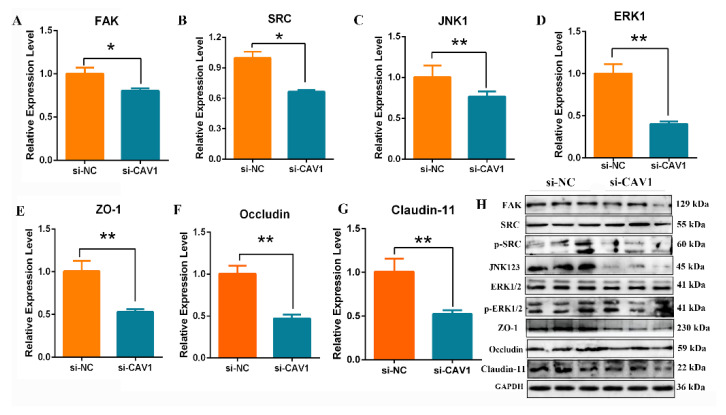
CAV1 suppression of TJs and BTBs of Sertoli cells. (**A**–**G**) The mRNA expression of *FAK, SRC, JNK, ERK, ZO-1, occludin* and *claudin-11* was detected by qRT-PCR after being transfected for siRNA-CAV1; values represent mean ± SD, *n* = 3. ** p* < 0.05, *** p* < 0.01. (**H**) The protein expression of FAK, SRC, p-SRC, JNK1, ERK1, p-ERK1, ZO-1, occludin and claudin-11 was detected using Western blotting, *n* = 3.

**Figure 9 animals-14-00183-f009:**
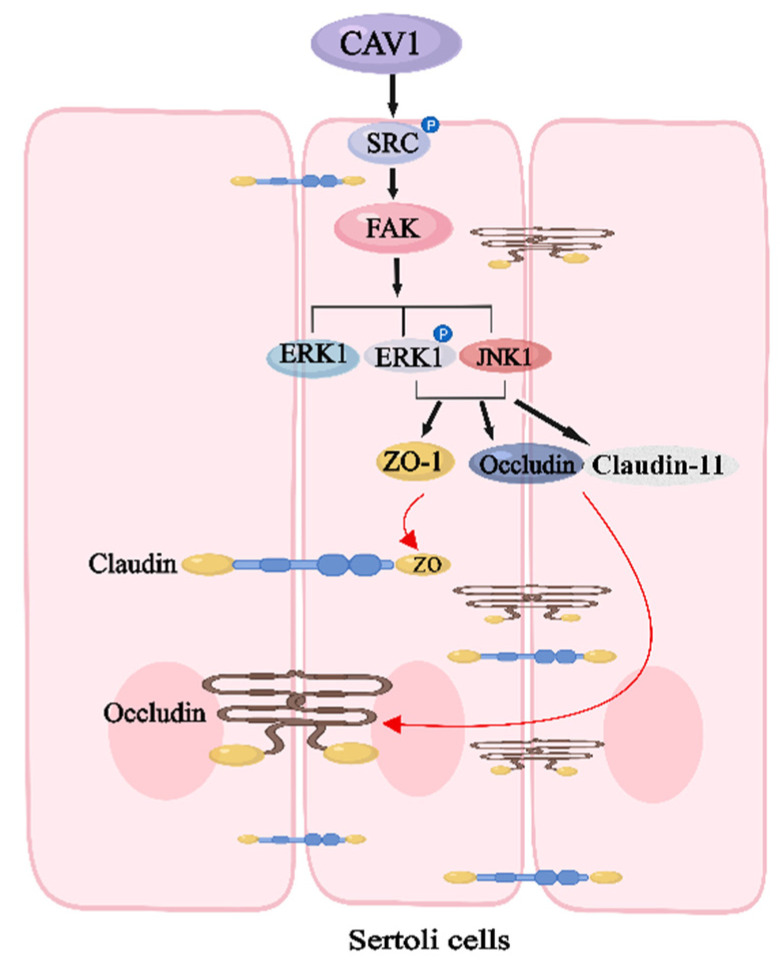
CAV1 regulates TJs and the integrity of BTBs of SCs via FAK/ERK signaling pathway. Abbreviations: CAV1: caveolin-1; SRC: SRC proto-oncogene; ERK: mitogen-activated protein kinase 1; JNK: c-Jun NH2-terminal kinase; ZO-1: TJP1 (ZO1) tight junction protein 1.

**Table 1 animals-14-00183-t001:** Information on primer sequences for PCR.

Name	Gene Sequence (5′–3′)	Tm (°C)
*ZO-1*	F: GAGGACAGTTACGACGAGG	59
	R: CAGGGTGACTTTGGTGGG	
*Occludin*	F: GAACGAGAAGCGACCGTATCC	59
	R: CTTGCTCTGCCCGCCTTG	
*CAV1*	F: GCAACATCTACAAGCCCAACA	56
	R: GCAGACAGCAAACGGTAAAAC	
*SRC*	F: CCTCAGGCATGGCTTATGT	54
	R: ACCCGTCCCTTTGTTGTG	
*FAK*	F: GACAGTTACAACGAGGGCGTCAA	58
	R: GGCGGGCAGAACAGGAATG	
*ERK1*	F: CAACACCACCTGCGACCTTA	56
	R: GGGATGGGGAGCCTAGAATA	
*TGFB1*	F: GGTGGAATACGGCAACAAA	55
	R: GTGGGCACTGAGGCGAAA	
*HOXD10*	F: TGGACAGACCCGAACAGAT	55
	R: GATGGGACCTCAGCAGAAAT	
*LCP2*	F: CAAGTCCAGCGGTTTCC	55
	R: GGGTGGTGGCTCGTAAT	
*GAPDH*	F: GCTGGTGCTGAGTATGTGGTG	58
	R: GCTGACAATCTTGAGGGTGTTG	
*CD27*	F: GGGAGGATGGTGTTGTCAGA	56
	R: TTCAGCATAAGGTAAGTGGGAG	
*JNK1*	F: GTTGACATTTGGTCAGTTGGG	58
	R: GGGAAAAGTACATCAGGGAAGA	
*ITGAM*	F: AGTCTGCCTCCAAGTTCGC	55
	R: TTCAGGGTCTCGCATTTCT	
*TLR4*	F: CTGCCTTCACTACAGGGACTT	56
	R: TGGGACACCACGACAATAAC	
*LPXN*	F: CCCTTCCGTTCCTGATGACA	56
	R: TGCGGCTGCTGAGGTTTTA	
*Claudin*	F: TATGGCTACGGGGCTTTAC	57
	R: AACTGTGAGCAGCAGGAGAAT	
*miR-150-x*	F: AGTGGTCGTATCCAGTGCG	60
*miR-1247-x*	F: CCCGGAGTCGTATCCAGTG	57
*miR-2713-z*	F: CGGGCATCATTGAAGGTC	60
*miR-12308-y*	F: TAAATCCCTGCTCTGACACG	56
*miR-2432-z*	F: TAGAGGTCGTCGTATCCAGTG	58
*miR-2431-y*	F: GTGCGAATACCTCGGACC	58
*novel-m0399-3p*	F: GAGCGTCGTATCCAGTGCG	62
*novel-m0230-3p*	F: AGCGTGAGCAGGAGCAGC	60
*U6*	R: CGAGGATGTGAAGACACCAAGAC	59

## Data Availability

All data are available in the article.

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
