# Peer review of "Caveolin 1 Regulates the Tight Junctions between Sertoli Cells and Promotes the Integrity of Blood–Testis Barrier in Yak via the FAK/ERK Signaling Pathway"

_animals, 2024, doi:10.3390/ani14020183_

Round 1
Reviewer 1 Report
Comments and Suggestions for Authors
In my opinion this is an interesting paper. I comment the authors on the quality of their work.
Line 39. I would change the sentence to read like this:
"Yaks are a unique bovine species endemic to high-altitude regions. Cryptorchidism is a common disease of the reproductive system and one of the main causes of infertility in male yaks."
Line 54: Remove "In a word" and start sentence with: BTB
Line 56: remove "amazing"
Line 78: Full stop required after "G proteins [15, 16]"
Line 121: Please describe this method in detail, or at least provide a reference where it has been described in case others look to replicate.
Line 373: "no" should be "not"
Line 390: remove "Clearly"
Line 391: Change "down-creased" to "decreased"
Line 408: Remove "And" - start the sentence with "In"
Line 417: Remove "about specific" and replace with "in"
Comments on the Quality of English Language
The English was quite good. Some very minor changes have been recommended.
Author Response
Dear reviewer:
We are grateful to you for your professional review work on our manuscript. ​Following your kind suggestions, we have made extensive corrections to our previous manuscript and we have answered all questions one by one, the detailed corrections are listed in the attachments.
Please do not bother to contact us if there is anything we can amend.. Thank you so much for your help.

Reviewer 2 Report
Comments and Suggestions for Authors
The topic has scientific relevance. This research can contribute and validate insights about CAV1 action in yak cryptorchidism. In general, the methods are appropriate for the objectives. The statistical analyses used are not the most adequate, for the few numbers of samples. The discussion touches on the main findings and their interpretation. In general, the topics chosen for discussion are adequate and interesting. However, the weaker sides of the manuscript the presentation which is not mature enough for publication.
Specific line comments:
L83-90: the introduction is quite verbose; I suggest moving this paragraph into the discussion.
L91-96: I believe this paragraph is not adequate for the introduction, it should be included in the conclusions.
L99: why so few samples? Statistically not very heavy in terms of number of samples. Also, what is the age of the animals from which you collected? Could it influence the data?
L:111 you should change the value “106” to “106”.
L188 (Table 1): has the efficiency of the primers been evaluated previously?
L218: Have the proteins been quantified previously? The volume could contain a different number of proteins, of consequence it should be the same for all to be able to be evaluated.
L216: what is the WB marker?
Figure 1: How could the non-specific bands in the detection of occludin present in this figure be justified? I would also insert the reference bar of the length of the immunohistochemistry image.
L332-341: it should be included in the materials and methods.
L342 : GAPDH in the first two samples is clearer, could this be due to the unquantified protein concentration?
Figure 6: CAV1 image, which band is evaluated: the first, the second or both? Is there any information in the literature about different forms of CAV1?
Figure 7: In the SRC graph in the WB figure there appear to be no real statistical differences.
Figure 7 and 8: some WB images are not ideal to analyse (pSRC, pERK1/2,zo-1), too much black background which could induce errors in quantification with ImageJ.
L:399-436: long-winded discussion, it is certainly interesting in terms of topics covered, but the thread of the discussion and the main point are lost. I would shorten the discussion.
Supplementary materials: why cut the marker part? It would be advisable not to cut the gels and membranes.
Comments on the Quality of English LanguageEnglish is quite clear.
Author Response

(The authors gave the same response as above.)

Reviewer 3 Report
Comments and Suggestions for Authors
The authors present a work in relation to elucidating some mechanisms that modulate the integrity of the blood-testicular barrier during the appearance of cryptorchidism in testis of male yaks, a unique bovine species, endemic to high altitude regions, a pathology that is considered one of the main causes of infertility in yaks. Using testicular tissue and cell cultures of isolated Sertoli cells, the expression of proteins that constitute the tight junctions of the testis, such as ZO-1 and occludin, were analyzed. Furthermore, using methodological approaches of RNA-seq and bioinformatics analysis, it was concluded that the cavelonin I protein, through the FAK/ERK signaling pathway, would be involved in the mechanisms that modulate tight junctions and therefore in the regulation of cryptorchidism in male yak
Comments and Suggestions for Authors::

Author Response

(The authors gave the same response as above.)

Reviewer 4 Report
Comments and Suggestions for Authors
This work is well designed and well conducted. Results are reliable and discussion is meaningful. However, some errors such as mistyping, clarity of figures et.al are still present in the manuscript. So, before the manuscript be suitable for publication, some minor revisions are still required. Note that the authors should carefully revise point by point.
Specific comments:
1. Please unify the format of nouns in the manuscript, such as "Western blot" and "Western blotting".
2. In Section 2.1. Animals and Sample Collection, only the details of how the samples were collected for subsequent molecular experiments are mentioned. While for the subsequent Section 2.5. Cell primary culture and identification, what does it have to do with tissue samples for primary cells? Please add the details in the appropriate location in this manuscript.
3. ​It is recommended that providing carrier enzyme loci and plasmid profiles (Section 2.4) in attachment for reviewing.
4. It is recommended that the labeling method of saliency in Figure 1B be unified with other figures.
5. The title “3.2. GO and KEGG analysis of different expressed target genes” is more like a description of a method and is suggested to be modified.
6. In Section 2.6. Cell transfection. For transfection of eukaryotic vectors, the conventional transfection reagent is Lipofectamine@2000 or Lipofectamine@3000 Reagent, why did the authors choose the not widely used PolyJetTM Reagent to transfect overexpression vectors? And is there any advantage? Please explain it.
7. ​Try to improve the resolution of the images before publication, for example, Figure 2B and Figure 2C are not clear.
8. ​Testis tissue is rich in cell types. It is recommended that the authors add descriptions of cell types in all immunofluorescence and immunohistochemical figures.
Author Response

(The authors gave the same response as above.)

Round 2
Reviewer 3 Report
Comments and Suggestions for Authors
Dear authors:
I have reviewed your responses to the revision of your paper... and based on them and the significant changes that were made, I agree with your modifications to the manuscript.
For this reason, I consider that this manuscript can be considered for publication in this journal.
Kind regards